# Optimization of Traction Magnetic Resonance Imaging to Improve Visibility of the Elbow Cartilage

**DOI:** 10.3390/diagnostics14060630

**Published:** 2024-03-16

**Authors:** Sho Kohyama, Kazuhiro Ikeda, Yoshikazu Okamoto, Naoyuki Ochiai, Yuichi Yoshii

**Affiliations:** 1Department of Orthopaedic Surgery, Kikkoman General Hospital, Noda 278-005, Chiba, Japan; sho_kohyama_1025@yahoo.co.jp (S.K.);; 2Department of Diagnostic and Interventional Radiology, Faculty of Medicine, University of Tsukuba, Tsukuba 305-8575, Ibaraki, Japan; 3Department of Orthopaedic Surgery, Tokyo Medical University Ibaraki Medical Center, Ami 300-0395, Ibaraki, Japan

**Keywords:** MRI, traction, elbow, articular cartilage

## Abstract

We previously reported that elbow magnetic resonance imaging (MRI) with 7 kg traction increases the joint space width of the radiocapitellar joint and improves articular cartilage visibility without arthrography. However, the optimal traction weight remains unclear. We assessed the effects of different traction weights on elbow MRI in 30 healthy volunteers. Elbow MRI was performed without traction and with 3, 5, and 7 kg axial tractions. The joint space width, humeral articular cartilage outline visibility, and intraprocedural pain/discomfort were evaluated. The joint and cartilage parameters were measured at the radiocapitellar joint and the lateral and medial thirds of the ulnohumeral joint. At the radiocapitellar joint, the joint space width increased significantly with traction. The cartilage outline visibility significantly increased with traction, with no significant differences among the traction weights. No significant result was observed at the lateral and medial thirds of the ulnohumeral joint. Pain and discomfort significantly increased as we used heavier traction weights. Elbow MRI with 3 kg traction showed sufficient effects similar to those observed with 7 kg traction with minimal pain and discomfort. There was no difference in the effect of traction between male and female participants. This procedure may enable enhanced visualization of intra-articular elbow injuries.

## 1. Introduction

Treatment of articular cartilage injuries requires precise evaluation of the articular cartilage to determine the appropriate strategies. Conditions that may trigger articular cartilage injuries include osteoarthritis of the elbow, capitellar osteochondritis dissecans, chondromalacia of the trochlea and trochlear notch [1,2,3,4,5,6], osteochondral fractures and cartilage injuries of the capitellum concomitant with posterolateral rotational instability [7], and radial head fractures [8]. Additionally, rheumatoid arthritis and other collagen diseases may also cause deterioration of the elbow function by destruction of the joint [9]. The elbow is anatomically complex in shape but relatively small, which sometimes makes it difficult to precisely evaluate the state of the articular cartilage using magnetic resonance imaging (MRI) [10]. In addition, the surface of the articular cartilage of the capitellum, where most cartilage injuries occur, is close to the facing radial head [11]. Consequently, because opposing articular cartilages have similar MRI signal intensities [11], it is difficult to adequately determine articular cartilage outlines for precise lesion evaluation [12]. In order to correctly diagnose the state of the articular cartilage condition and its pathology, it is essential to obtain precise intraarticular images. Moreover, the degree of synovitis, bone edema, and cartilage destruction are crucial factors in planning treatment strategies for rheumatoid arthritis patients [9]. Better MR images facilitate more tailored treatment by precisely evaluating the lesion.

In previous research on this conundrum, we reported that 7 kg of axial traction during elbow MRI significantly increased the joint space width (JSW) of the radiocapitellar joint (RCJ) and improved articular cartilage visibility without the use of a contrast medium [13]. Lee et al. used 5 kg for female participants and 7 kg for male participants in traction MR arthrography of the elbow [12]. However, they did not provide supportive evidence regarding how they decided the traction weight. More evidence is required to determine the ideal traction weight for traction MRI of the elbow. Recent studies have applied axial traction to the thumb carpometacarpal joint [14], knee [15], and shoulder [16]. These studies found 5 kg of axial traction to be safe and sufficient for visualizing the thumb carpometacarpal joint and knee articular cartilage without arthrography, whereas 4 kg was sufficient to demonstrate morphological changes in the shoulder.

In the present study, we hypothesized that a weight of less than 7 kg would have a similar effect on traction MRI of the elbow. Accordingly, this study aimed to evaluate the effects of traction on MRI assessment of the elbow using different traction weights in healthy volunteers. We also aimed to evaluate sex differences in the effect of traction.

## 2. Materials and Methods

### 2.1. Recruitment of Volunteers

The institutional review board of Kikkoman General Hospital approved this study (no. KC-H24), and the trial was conducted in accordance with the Declaration of Helsinki (revised in 2013). We enrolled 30 healthy volunteers, 15 from male and 15 female, from the hospital’s working staff without any past elbow injury or any current symptoms associated with the elbow. The first author recruited volunteers by announcing the study after obtaining approval from the institutional review board. Written informed consent was obtained from all individual volunteers included in the study; all underwent clinical examination by the first author, whose career as an elbow surgeon spans more than 15 years. The volunteers were tested for range of motion, instability, and tenderness on the lateral and medial sides of the elbow. Volunteers were to be excluded if abnormal findings were found, but none exhibited abnormal findings; thus, all volunteers were included in the study.

### 2.2. Image Acquisition

We performed MRI between April 2021 and November 2023 using a 1.5-Tesla system (Achieva, Philips Medical Systems©, Best, The Netherlands) and an eight-channel coil (Philips Medical Systems©, Best, The Netherlands). We used the same sequence and parameters as used in a previous study: a three-dimensional T1-fast echo with water excitation for cartilage (WATS-c), field of view, 130 × 130 × 60 mm; matrix 256 × 195; slice thickness, 0.4 mm; slice gap, 0.4 mm; time to repeat, 20 ms; flip angle, 25°; and echo time, 8.4 ms [12]. Investigations were conducted on the right upper extremity, the dominant arm of all volunteers. The duration for acquiring each image was 3 min and 52 s. A few minutes were needed to adjust the traction weight between imaging procedures; therefore, each volunteer underwent image acquisition for approximately 20 min. If a volunteer moved involuntarily during image acquisition, the acquisition sequence was repeated to avoid motion artifacts.

### 2.3. Traction MRI

We initially performed MRI without traction before performing the traction MRI. First, we wrapped the right wrist of each volunteer with a sponge and a rope. Next, we attached a weight to the other end of the rope, which was hung over the edge of the MR table using a pulley system. The traction system was the same as previously described in [12] (Figure 1). We initiated axial traction with 3 kg and then increased the weight to 5 kg and then 7 kg.

### 2.4. Image Analysis

From the MR images, we evaluated the JSW, humeral articular cartilage outline visibility (HACOV), and intra- and inter-rater reliability for assessing the JSW and HACOV. As most articular cartilage injuries of the elbow occur on humeral articular surfaces [5,6,7,8], we only investigated the humeral articular cartilage. We defined the JSW as the distance from the humeral articular cartilage surface to the facing articular surfaces of the assessed joint.

Two examiners, both elbow surgeons with over 10 years of clinical experience, independently evaluated all MR images under the supervision of a musculoskeletal radiologist (Y.O.) with more than 20 years of clinical experience. We used Osirix MD (version 14.0 Pixmeo©, Bernex, Switzerland) to interpret the images and obtain multiplanar reconstructed images. The first author reconstructed all the sagittal and coronal images parallel to the longitudinal axis of the humerus. The examiners freely enlarged and adjusted the grayscale contrast of the images during assessments for optimal visualization of the structures. All MR images were randomly numbered to reduce potential examiner bias. Even with MRI, examiner-related measurement errors may occur. Therefore, both examiners measured each parameter twice to evaluate intraobserver correlations. The second assessment was conducted at least two weeks after the first assessment.

### 2.5. Measurement of JSW

We measured the JSW at the RCJ, the lateral third of the ulnohumeral joint (LUHJ), and the medial third of the ulnohumeral joint (MUHJ). We used the same JSW measurement point definitions and HACOV grading procedures as those described in a previous study [12] (Figure 2), and the examiners understood and adhered to the definitions to ensure consistency in the study. Specifically, on the coronal image in which the radial head was depicted as the largest, we determined the midpoint of the radial head and the lateral and medial thirds of the ulna (Figure 2a). We measured the joint space widths on a sagittal image passing through each point. On a sagittal image passing through the midpoint of the radial head on the coronal image, we again determined the midpoint of the radial head, defined as the center of the radial head. We measured the joint space width of the RCJ along the vertical line extending proximally from the center of the radial head (Figure 2b).

On the sagittal image passing through the lateral and medial thirds of the ulna, we determined the bisector of the humerus that is horizontal to the longitudinal axis of the humerus. Subsequently, we measured the joint space width at the LUHJ and MUHJ along the line extending distally from the bisector (Figure 2c,d). 

### 2.6. Assessment of HACOV

On the sagittal images with which we measured the JSWs, we graded the HACOV using the same three-point scale used in a previous study [13]. Grading was performed on the sagittal image with measured joint space widths. Visibility was graded as poor when <50% of the humeral articular cartilage outline was visible in the range facing the opposing articular cartilage (Figure 3a). When ≥50% but <100% of the humeral articular cartilage outline was visible, visibility was graded as intermediate (Figure 3b). Visibility was graded as complete when 100% of the humeral articular cartilage outline was visible (Figure 3c).

### 2.7. Evaluation of Pain and Discomfort 

We further evaluated pain and discomfort during traction MRI. Each volunteer was asked to complete the visual analog scale questionnaire immediately after every MRI session to determine whether axial traction was used. Each volunteer rated their pain and discomfort using visual analog scale scores ranging from 0 to 10. The results of the questionnaire were then aggregated.

### 2.8. Statistical Analyses

We used SPSS Statistics version 29 (IBM, Armonk, NY, USA) for statistical analyses. We used the intraclass correlation coefficient (г) to calculate the intra- and inter-observer correlation of JSW assessments. For these correlations, we graded the г value as follows: >0.8, excellent correlation; 0.6–0.8, good correlation; 0.4–0.6, moderate correlation; 0.2–0.4, fair correlation; and <0.2, poor correlation [17]. Finally, we used Cohen’s kappa (k) statistic to assess the intra- and inter-observer reliabilities of the HACOV [18]. For these assessments, we graded the k value as follows: >0.90, almost perfect agreement; 0.80–0.90, strong agreement; 0.60–0.79, moderate agreement; 0.40–0.59, weak agreement; 0.21–0.39, minimal agreement; and 0–0.20, no agreement [19]. 

Normal distribution was assessed using the Shapiro–Wilk test, and only the data of the JSW of RCJ followed a normal distribution; others did not. Therefore, we used the paired *t*-test for JSW of the RCJ and Wilcoxon signed-rank test to assess the statistical significance of the differences in JSW other than RCJ, HACOV, pain, and discomfort assessments for each examination. 

We also evaluated sex differences in the effect of axial traction by comparing the data obtained from female and male volunteers. When the data were divided according to participant sex into female and male groups, the data of the JSW of the RCJ, LUHJ, and MUHJ followed a normal distribution, but the data of the HACOV did not. Therefore, we used the paired *t*-test to evaluate JSW and Wilcoxon signed-rank test to assess the statistical significance of the differences in the rest of the parameters. 

Statistical significance was set at a *p*-value < 0.05 for all statistical analyses. 

## 3. Results

In this study, among 30 volunteers, we equally enrolled 15 male participants and 15 female participants; the mean age was 34.6 ± 1.6 (range 22–49) years. None of the volunteers had a history of elbow injury. The demographic data of each volunteer are presented in Table 1.

The intraobserver correlations of the JSW of examiner 1 were 0.99 for RCJ, 0.95 for LUHJ, and 0.93 for MUHJ. For examiner 2, the intraobserver correlations of the JSW were 0.98 for RCJ, 0.85 for LUHJ, and 0.86 for MUHJ. The inter-observer correlations of the JSWs for RCJ, LUHJ, and MUHJ were 0.99, 0.98, and 0.97, respectively. The correlations were excellent for all measurements conducted by both examiners. 

The intraobserver correlations of the HACOV of examiner 1 were 0.92 for RCJ, 0.94 for LUHJ, and 0.95 for MUHJ. For examiner 2, the intraobserver correlations of the JSW were 0.91 for RCJ, 0.86 for LUHJ, and 1.0 for MUHJ. The inter-observer correlations of the HACOVs for RCJ, LUHJ, and MUHJ were 0.97, 0.88, and 0.83, respectively. The agreements were excellent for all measurements conducted by both examiners. 

These results indicate the robust reproducibility of our measurement criteria in this study. Upon these, we present the first measurement performed by examiner 1 as representative data in the following section. Prior to the data presentation, we present representative MR images of the RCJ, LUHJ, and MUHJ at every setting in Figure 4 to illustrate the effectiveness of the axial traction.

In the RCJ, the JSW significantly increased with the application of traction. However, there were no significant differences among the traction weights used: *p* < 0.01 for 0 kg vs. 3 kg, *p* < 0.01 for 0 kg vs. 5 kg, *p* < 0.01 for 0 kg vs. 7 kg, *p* = 0.13 for 3 kg vs. 5 kg, *p* = 0.06 for 3 kg vs. 7 kg, and *p* = 0.63 for 5 kg vs. 7 kg (Figure 5). The HACOV also significantly increased with the application of traction. The rate of RCJ assessments evaluated as complete on conventional MRI was 27% (8/30 assessments). The rate was 83% (25/30 assessments, *p* < 0.01) for 3 kg traction, 87% (26/30 assessments, *p* < 0.01) for 5 kg traction, and 93% (28/30 assessments, *p* < 0.01) for 7 kg traction. However, no significant differences were observed between the different traction weights used: *p* = 0.73 for 3 kg vs. 5 kg, *p* = 0.248 for 3 kg vs. 7 kg, and *p* = 0.410 for 5 kg vs. 7 kg (Figure 6).

At the LUHJ, the JSW did not show a significant difference with the application of traction: *p* = 0.51 for 0 kg vs. 3 kg, *p* = 0.14 for 0 kg vs. 5 kg, *p* = 0.23 for 0 kg vs. 7 kg, *p* = 0.26 for 3 kg vs. 5 kg, *p* = 0.55 for 3 kg vs. 7 kg, and *p* = 0.62 for 5 kg vs. 7 kg (Figure 7). The HACOV showed no significant improvement with the application of traction either: *p* = 0.17 for 0 kg vs. 3 kg, *p* = 0.06 for 0 kg vs. 5 kg, *p* = 0.06 for 0 kg vs. 7 kg, *p* = 0.63 for 3 kg vs. 5 kg, *p* = 0.55 for 3 kg vs. 7 kg, and *p* = 0.89 for 5 kg vs. 7 kg (Figure 8).

At the MUHJ, the JSW did not show a significant difference with the application of traction: *p* = 0.54 for 0 kg vs. 3 kg, *p* = 0.81 for 0 kg vs. 5 kg, *p* = 0.57 for 0 kg vs. 7 kg, *p* = 0.65 for 3 kg vs. 5 kg, *p* = 0.92 for 3 kg vs. 7 kg, and *p* = 0.76 for 5 kg vs. 7 kg (Figure 9). No volunteer was graded complete on the HACOV, and it showed no significant improvement with the application of traction: *p* = 0.31 for 0 kg vs. 3 kg, *p* = 0.31 for 0 kg vs. 5 kg, *p* = 0.17 for 0 kg vs. 7 kg, *p* = 1.0 for 3 kg vs. 5 kg, *p* = 0.69 for 3 kg vs. 7 kg, and *p* = 0.69 for 5 kg vs. 7 kg (Figure 10).

Pain and discomfort significantly increased with traction, and the volunteers scored significantly higher VAS scores as we used heavier traction weights; *p* < 0.01 was used for all combinations (Figure 11). 

There were no sex differences in the effect of traction on JSW, HACOV, pain, and discomfort. At the RCJ, *p*-values for the assessment of JSW were 0.64 for 0 kg, 0.19 for 3 kg, 0.61 for 5 kg, and 0.36 for 7 kg. For HACOV, *p*-values were 0.84, 0.48, 0.22, and 0.89, respectively. At the LUHJ, *p*-values were 0.08, 0.19, 0.37, and 0.15 for JSW, and 0.35, 0.87, 0.85, and 0.71 for HACOV, respectively. At the MUHJ, *p*-values were 0.75, 0.61, 0.34, and 0.18 for JSW, and 0.35, 0.63, 0.09, and 0.36 for HACOV, respectively. For pain, *p*-values were 0.52, 0.41, 0.69, and 0.62, respectively. For discomfort, *p*-values were 0.12, 0.48, 0.58, and 0.6, respectively. We present the data of the JSW of the RCJ as a representative figure (Figure 12).

## 4. Discussion

As proposed in a previous study [13], traction was demonstrated to be effective in the articular cartilage evaluation of the elbow in the present study, particularly in the evaluation of the RCJ. The traction significantly widened the space of the RCJ, regardless of the weight used. However, there were no significant differences between the traction weights we used: 3 kg, 5 kg, and 7 kg. This suggested that the joint-space-widening effect of traction does not increase linearly. There were also no significant differences in the improvement in cartilage outline visibility of the RCJ with traction between the traction weights used. These results indicate that the effect of 3 kg traction is comparable to that of 7 kg traction in terms of improvement in cartilage outline visibility.

The inter- and intra-observer correlations of the JSWs and HACOVs were excellent at all measurement points, suggesting that the assessment procedure in this study, including the supervision of an experienced radiologist, is highly reproducible and reliable.

The RCJ bears 57% of the load across the elbow [20], with a greater load depending on the activity or in the event of a fall [21,22]. The force transmission at the RCJ is the greatest when the elbow is extended [23]. Morrey reported in his biomechanical study that the mechanical load through the radial head was 0.5 kg when the forearm was in supination and nearly 2.5 kg when the forearm was in pronation [23]. In the present study, the forearm position was not specified. Therefore, we believe it is reasonable to apply traction with a weight of 3 kg to widen the joint space and evaluate the outline of the articular cartilage of the elbow, particularly at the RCJ. Additionally, osteochondritis dissecans of the elbow mainly occur in the RCJ [1,2,3,4,5]. When considering articular surface reconstruction in severe cases, obtaining preoperatibe images with traction MRI would be of great advantage.

We made another interesting observation about the LUHJ, although there were no significant differences: the JSWs slightly increased until 5 kg traction, but when a traction weight of 7 kg was applied, the JSW of the LUHJ narrowed compared with that with 5 kg of traction. Anatomically, owing to the morphology of the distal humerus, the elbow shows a slight valgus angulation in extension, described as the carrying angle [24]. Given that repetitive valgus stress is applied to the elbow during daily activities, the medial collateral ligament is the strongest ligament of the elbow [24,25], turning it into a loose hinge joint with screw displacement movement and the medial epicondyle as the vertex [21]. When 7 kg of traction was applied, the RCJ was overtractioned, which may have changed the alignment of the elbow from anatomical valgus to varus, thereby narrowing the joint space of the LUHJ. This finding suggests that 7 kg of traction is too high for the LUHJ. However, although the JSW of the LUHJ increased with 5 kg of traction, we did not observe an improvement in the HACOV. Therefore, it may still be difficult to thoroughly evaluate the articular cartilage condition in cases of severe osteoarthritis with cartilage damage in the ulnohumeral joint. Further improvement is necessary to overcome this problem.

The mean pain and discomfort values significantly increased with the application of traction, and both increased when heavier weights were used. Notably, the mean pain and discomfort scores were not zero with conventional MRI. Although our previous results suggested that 7 kg of traction is safe (mean visual analog scale score of 2.0 for both pain and discomfort), this study yielded higher scores: 3.9 for pain and 4.7 for discomfort. This divergence is probably due to differences in volunteer characteristics between this study and the previous study [13]. Based on these results, we do not recommend using 7 kg traction in daily practice.

Considering sex differences, we observed no significant differences in the effect of traction on JSW, HACOV, pain, and discomfort between the groupd of female and male participants. Our results are inconsistent with those of the study by Lee et al., which recommends 5 kg for female patients and 7 kg for male patients [12]. We conclude that there is no need to use different traction weights considering sex, and a traction weight of 3 kg leads to a comparative effect to heavier weights.

This study has some limitations. First, the small sample size limits the power of the study. The study focused on healthy volunteers from a specific hospital’s working staff aged 20–40 years, which might not fully represent the broader population, especially individuals with elbow injuries or conditions. The findings might have limited applicability to different demographics or patient groups. Since the aim of the present study was to determine the appropriate traction weight, we considered it inappropriate to include patients in the study population. We plan to prove the effect of traction on actual patients in future studies. Second, the study was conducted at a single institution, which could introduce biases related to participant selection, imaging protocols, and other factors specific to that center. Multi-center studies could provide more robust and generalizable results. Third, we did not consider the body sizes of the volunteers. The effects of axial traction may differ among individuals with different body weights or muscle volumes. In this study, axial traction of 3 kg was sufficient for some volunteers for complete visualization of the articular cartilage outline, while others required heavier traction weights. Fourth, further investigations in children are needed, as some articular cartilage injuries, such as osteochondritis dissecans, occur predominantly in adolescents. In the future, we aim to create specific criteria for traction MRI based on the body size, age, and sex of each patient.

## 5. Conclusions

Traction MRI increased the JSW of the RCJ and improved the HACOV. Considering the observed effects and the invasive aspects of traction, a traction weight of 3 kg was considered appropriate for MRI evaluation of the elbow. There were no differences in the effect of axial traction between the sexes. Further investigation is required to assess its use in patients with elbow symptoms.

## Figures and Tables

**Figure 1 diagnostics-14-00630-f001:**
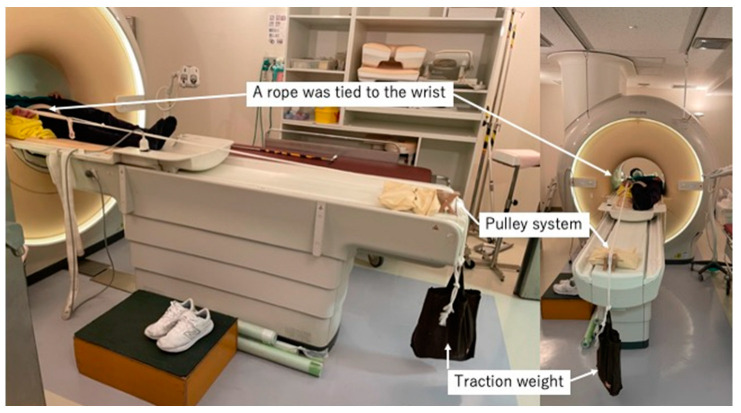
Overview of the traction system. A rope was tied to the volunteer’s wrist, and a weight was attached to the other end of the rope to hang over the edge of the table using a pulley system.

**Figure 2 diagnostics-14-00630-f002:**
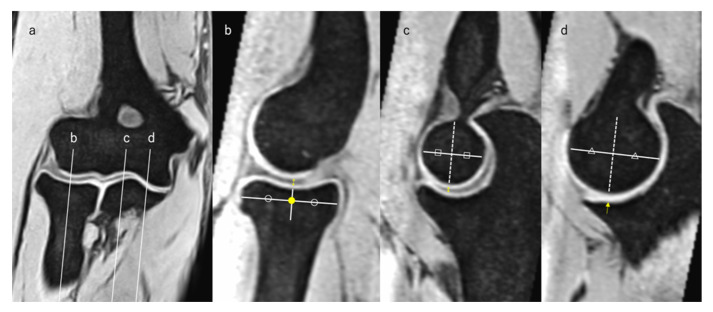
Definition of the measurement points. (**a**) The coronal image in which the radial head was depicted as the largest. Lines b, c, and d represent the midpoint of the radial head and the lateral and medial thirds of the ulna, respectively. (**b**) Measurement point of the RCJ. The yellow dot represents the center of the radial head. The double-headed arrow is the joint space width we measured. (**c**) Measurement point of the LUHJ. The dotted line is the bisector of the humerus, and the double-headed arrow is the joint space width we measured. (**d**) Measurement point of the MUHJ. The dotted line is the bisector of the humerus, and the arrow is the joint space width we measured. LUHJ—lateral third of the ulnohumeral joint; MUHJ—medial third of the ulnohumeral joint. On each figures, the same shapes (circles, squares and triangles) represents the same length, respectively.

**Figure 3 diagnostics-14-00630-f003:**
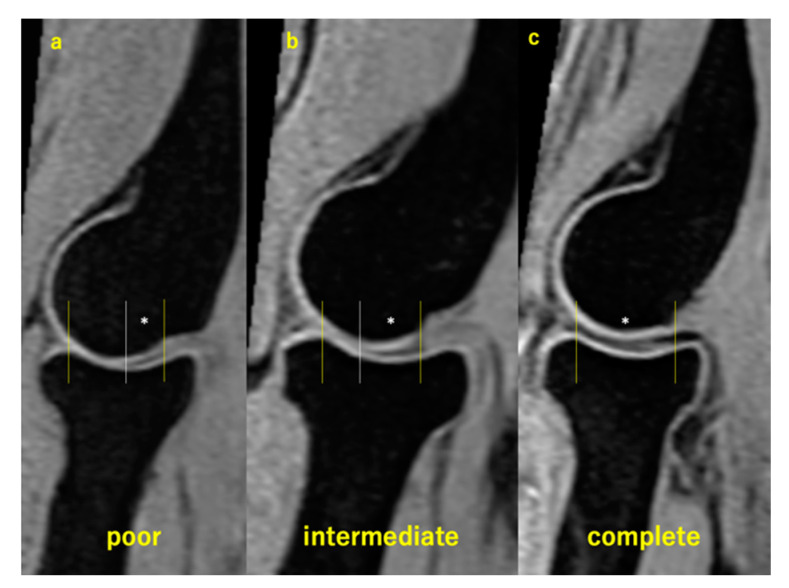
Grading of the humeral articular cartilage visibility. The area between yellow lines represents the range where the humerus faces the opposing articular cartilage. The white lines pass the point where opposing cartilages touch each other. The area with * represents where the humeral articular cartilage outline was visible. (**a**) Poor visibility; (**b**) intermediate visibility; (**c**) complete visibility.

**Figure 4 diagnostics-14-00630-f004:**
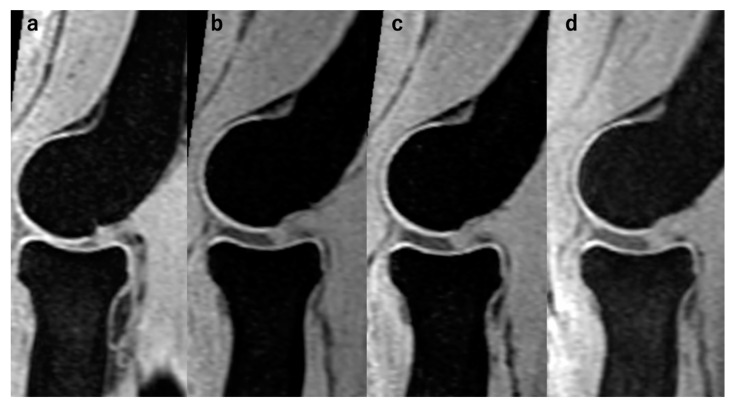
Representative MR images on the RCJ. The presented case is case 5. The JSW had been distracted by the application of axial traction. (**a**) Without traction; (**b**) with traction of 3 kg; (**c**) with traction of 5 kg; (**d**) with traction of 7 kg.

**Figure 5 diagnostics-14-00630-f005:**
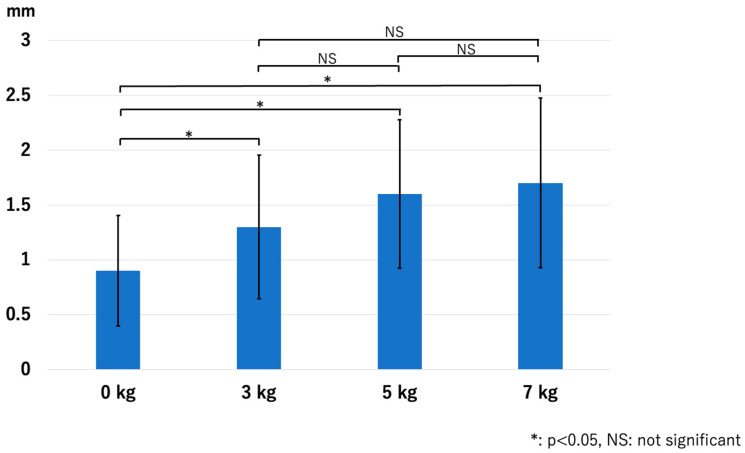
The joint space width at the RCJ. Joint space width increased significantly with traction application regardless of the traction weights used. RCJ—radiocapitellar joint.

**Figure 6 diagnostics-14-00630-f006:**
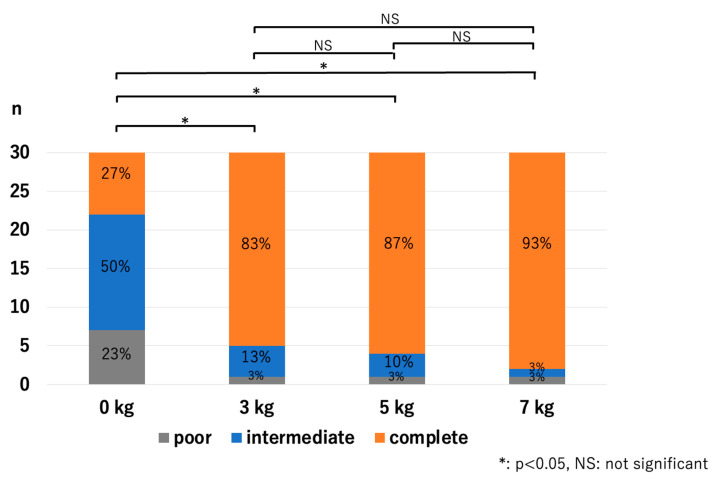
Cartilage outline visibilities at the RCJ. The application of traction significantly improved the visibility of the cartilage outlines; however, there were no significant differences in effect between the different traction weights (RCJ—radiocapitellar joint).

**Figure 7 diagnostics-14-00630-f007:**
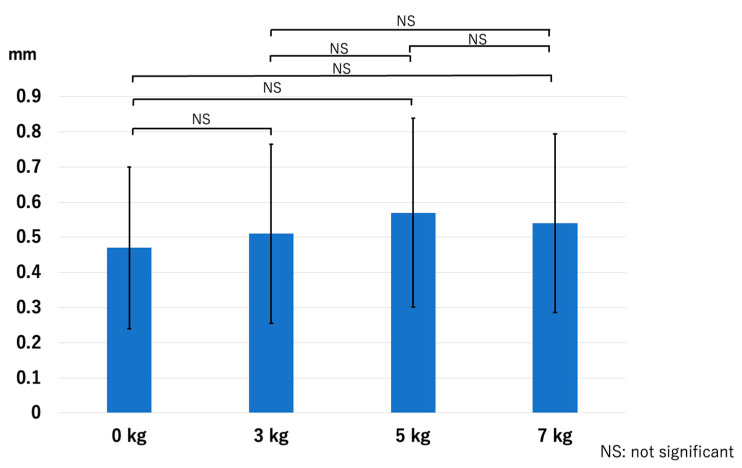
The joint space width at the LUHJ. Compared with the application of no traction, the joint space width showed no significant increase with the application of traction (LUHJ—lateral third of the ulnohumeral joint).

**Figure 8 diagnostics-14-00630-f008:**
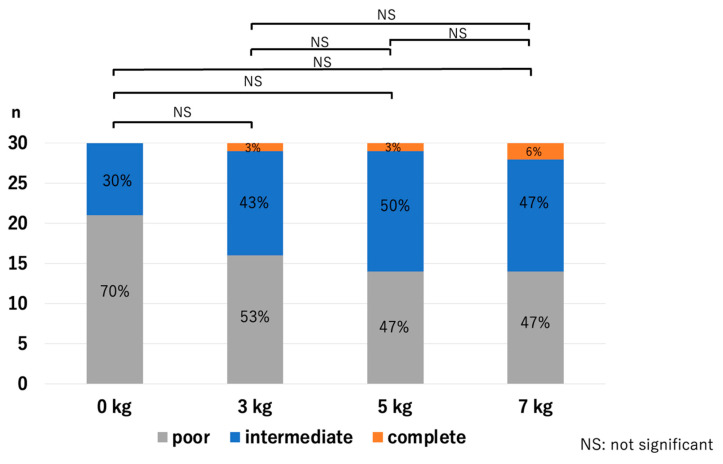
Cartilage outline visibilities at the LUHJ. Cartilage outline visibility did not improve with traction (LUHJ—lateral third of the ulnohumeral joint).

**Figure 9 diagnostics-14-00630-f009:**
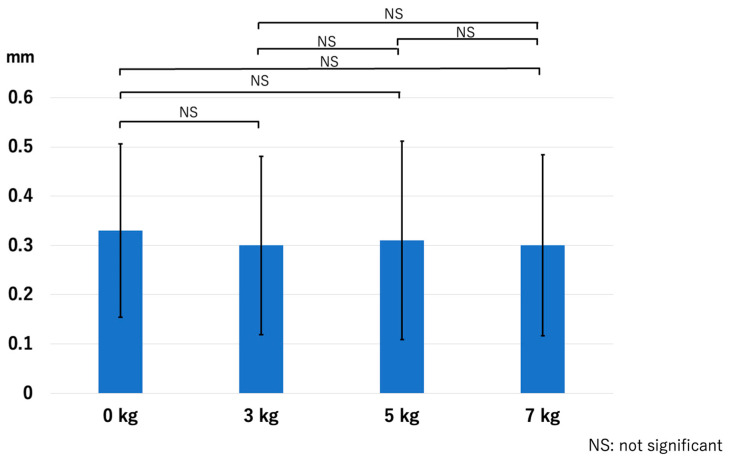
The joint space width at the MUHJ. No increase in joint space width at the MUHJ was observed, and the only significant difference observed was between no traction and 5 kg of traction in measurement 2-2 (MUHJ—medial third of the ulnohumeral joint).

**Figure 10 diagnostics-14-00630-f010:**
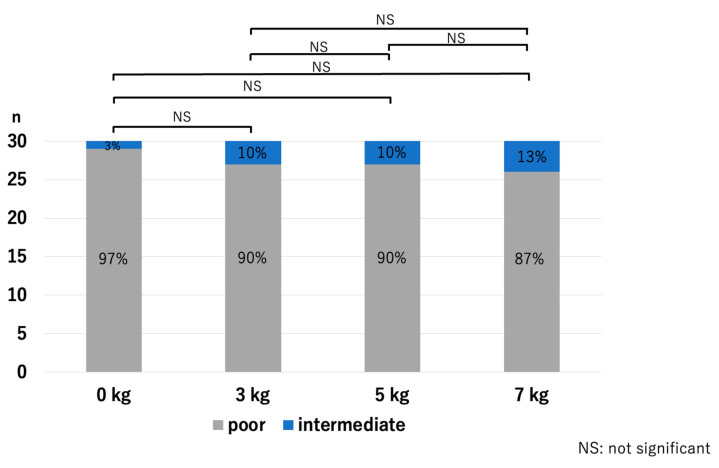
Cartilage outline visibilities at the MUHJ. There was no complete grade, and the cartilage outline visibility did not improve with traction (MUHJ—medial third of the ulnohumeral joint).

**Figure 11 diagnostics-14-00630-f011:**
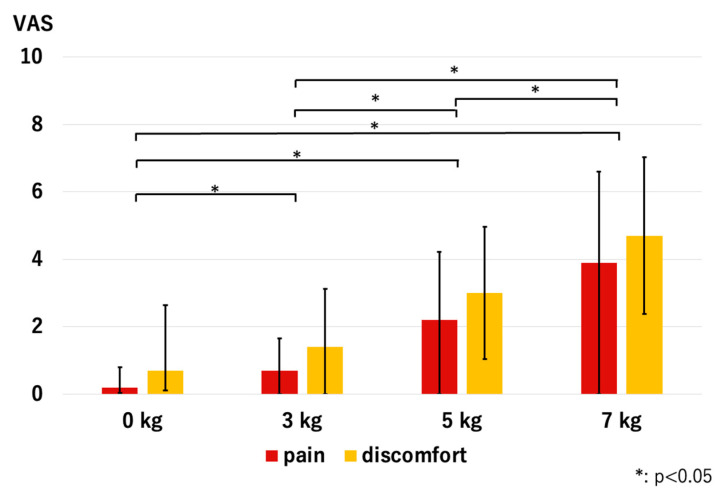
Pain and discomfort values reported by volunteers. Scores significantly increased as we used heavier traction weights (VAS—visual analog scale).

**Figure 12 diagnostics-14-00630-f012:**
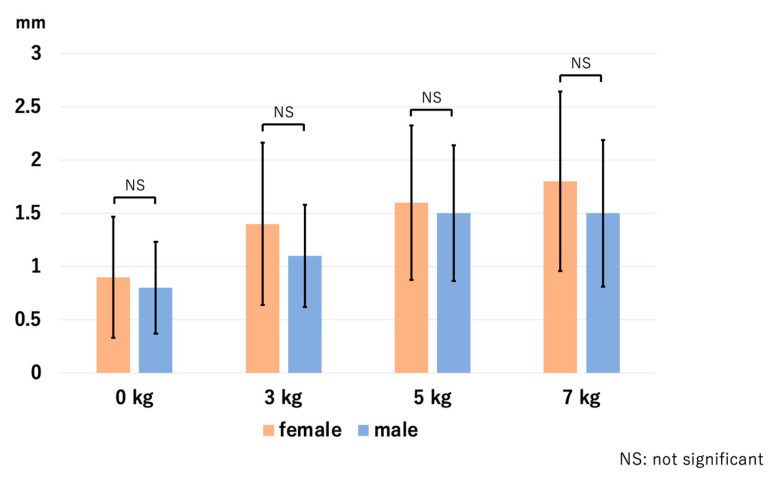
Sex differences in the JSW at the RCJ. There were no statistical differences between the two groups. JSW—joint space width; RCJ—radiocapitellar joint.

**Table 1 diagnostics-14-00630-t001:** Demographic data of the study participants.

Volunteer	Sex	Age (Years)	Volunteer	Sex	Age (Years)
1	Female	22	16	Male	23
2	Female	23	17	Male	23
3	Female	25	18	Male	26
4	Female	28	19	Male	26
5	Female	29	20	Male	28
6	Female	30	21	Male	32
7	Female	32	22	Male	33
8	Female	33	23	Male	34
9	Female	33	24	Male	38
10	Female	34	25	Male	38
11	Female	43	26	Male	41
12	Female	43	27	Male	43
13	Female	45	28	Male	45
14	Female	46	29	Male	46
15	Female	48	30	Male	49

## Data Availability

Data are contained within the article.

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
