# Peer review of "Optimization of Traction Magnetic Resonance Imaging to Improve Visibility of the Elbow Cartilage"

_diagnostics, 2024, doi:10.3390/diagnostics14060630_

Round 1

Reviewer 1 Report

Comments and Suggestions for Authors

The paper is very interesting and well structured, and the authors have expressed the limitations of the study very well.

Perhaps it would be useful to better specify in the introduction the usefulness of the paper not only to improve surgical access but also to facilitate tailored therapy, as it might be in the case of cartilage damage in rheumatologic diseases.

Author Response

The paper is very interesting and well structured, and the authors have expressed the limitations of the study very well.

Perhaps it would be useful to better specify in the introduction the usefulness of the paper not only to improve surgical access but also to facilitate tailored therapy, as it might be in the case of cartilage damage in rheumatologic diseases.

Response:

Thank you for your thoughtful comment. We have added statements in lines 37-38 and 49-52 to clarify the technique's usefulness in rheumatoid arthritis patients. The degree of synovitis, cartilage destruction, and bone edema are considered important predicting factors of rheumatoid arthritis. We believe our technique provides better images for evaluating these factors.

Reviewer 2 Report

Comments and Suggestions for Authors

The authors explore an interesting area of diagnostics using MRI. They proposed ways to increase the effectiveness of this analysis in elbow imaging.

The article is well structured and presents a large amount of data. However, MRI images from different experimental settings are not presented. I suggest that the authors add such a figure with representative MRI images to better understand how effective the proposed modifications are.

Author Response

The authors explore an interesting area of diagnostics using MRI. They proposed ways to increase the effectiveness of this analysis in elbow imaging.

The article is well structured and presents a large amount of data. However, MRI images from different experimental settings are not presented. I suggest that the authors add such a figure with representative MRI images to better understand how effective the proposed modifications are.

Response:

Thank you for your thoughtful comment. We have added a Figure 4 to present the effect of each traction weight. Since a noticeable effect was observed only on the RCJ, we presented the sagittal images of the RCJ. In the result section, we added the legends in lines 241-245.